# Effect of Food Matrix and Administration Timing on the Survival of *Lactobacillus rhamnosus* GG During In Vitro Gastrointestinal Digestion

**DOI:** 10.3390/foods14173076

**Published:** 2025-09-01

**Authors:** Junyan Wang, Peng Wu, Xiao Dong Chen, Aibing Yu, Sushil Dhital

**Affiliations:** 1Department of Chemical Engineering, Monash University, Clayton, VIC 3800, Australia; junyan.wang@monash.edu (J.W.); aibing.yu@monash.edu (A.Y.); 2School of Chemical and Environmental Engineering, College of Chemistry, Chemical Engineering and Material Science, Soochow University, Suzhou 215123, China; xdchen@mail.suda.edu.cn

**Keywords:** *Lactobacillus rhamnosus* GG, probiotic, in vitro digestion, viability, food matrix

## Abstract

Probiotics’ potential to enhance gut health is often limited by their poor survival during gastrointestinal (GI) transit, a challenge influenced by the composition and timing of co-ingested foods. Addressing the lack of dietary guidelines for optimal probiotic administration, this in vitro study examines how consuming *Lactobacillus rhamnosus* GG (LGG) with different foods at varying timings affects bacterial survival during simulated digestion. The results showed that simultaneous intake with durum wheat pasta or soy milk improved bacterial viability compared to standalone probiotics. The pasta outperformed the soy milk, yielding higher viable counts (5.92–6.38 vs. 4.93–5.39 log CFU/g) due to greater buffering capacity. Timing of administration also played a critical role: consuming probiotics with (5.39–5.92 log CFU/g) or after a meal (5.19–6.38 log CFU/g) enhanced viability compared to an empty-stomach scenario (4.93–6.04 log CFU/g). Additionally, LGG co-ingestion facilitated starch and protein digestion, increasing the pasta starch digestibility from 84.80% to 89.00% and the soy milk protein digestibility from 78.00% to 80.00%, suggesting synergistic bacteria–food interactions between the probiotic and food matrix. These findings emphasize the importance of food matrix selection and administration timing in optimizing probiotic efficacy. The study provides practical insights for healthcare professionals and consumers, advocating for meal-aligned probiotic intake with buffering-rich foods like pasta to maximize viability.

## 1. Introduction

The human gastrointestinal (GI) tract is an intricate and dynamic ecosystem where food, microbiota, and host interactions converge to impact overall health [1]. This complexity emphasizes the diverse roles the microbiome plays in energy balance, nutrient metabolism, and immune function [2]. Among the many interventions proposed to modulate this complex system, the administration of probiotics has gained considerable attention. Probiotics are delivered by fermented dairy products such as cheese and yogurt in the food industry, or directly in the form of capsules or powders in the nutritional supplement market [3].

Ingested along with food, probiotics encounter varying physicochemical conditions that can significantly influence their viability. The presence of digestive enzymes, low pH of the stomach, antimicrobial activity of bile salts, and peristaltic elimination in the human GI system exert a detrimental impact on the survival of bacterial cells [4]. Therefore, surviving the harsh GI environment is another challenge encountered by probiotics, apart from the technological stress they face during drying and processing as reported in our previous studies [5,6]. The efficacy of probiotics is contingent upon their survival through the GI tract and subsequent ability to maintain viability in the intestine. In order to fulfill their beneficial role and confer a health-promoting effect on the host, they should tolerate transit through the GI tract and reach the target site in the intestine alive and in adequate numbers to enable colonization and proliferation [7,8]. However, the majority of probiotics lack the ability to survive in large quantities through the GI passage [9].

The metabolic state and adaptability of bacteria ingested into the digestive tract are likely affected by the delivery method used [10]. Food is usually employed as their carrier matrix. However, the impact of the food matrix consumed simultaneously with probiotics is often neglected. Specific food matrices are expected to protect probiotic microorganisms during the GI tract passage [3,11,12]. Furthermore, the degree of protection conferred may depend on the composition of the carrier matrices [13]. For example, some studies suggest that the survival of probiotics is enhanced when consumed in dairy products such as milk, yogurts, cheese, and ice cream [10,14,15,16]. Moreover, the digestion rate of liquid foods was reported to be faster than that of solid foods, leading to reduced contact time of probiotics with stressors such as stomach acid [17]. Additionally, there is the potential for consumer confusion because sometimes they are instructed to take the probiotic supplements on an empty stomach, and sometimes they are advised to have them either with meals or before or after meals [3,18]. Although numerous studies have investigated the effects of different food matrices on probiotic viability [12,19,20], there remains a lack of consensus regarding the optimal timing for probiotic consumption relative to meals. As a protective vehicle that enhances probiotic viability, the timing of food intake relative to probiotic administration may be a critical determinant of the survival of probiotics due to potential buffering capacity.

Therefore, the objective of this study is to investigate how different food matrix compositions affect the survival of *Lactobacillus rhamnosus* GG (LGG) during simulated digestion, and to understand the relationship between bacterial survival and the timing of probiotic administration relative to mealtime. The ultimate aim is to determine the most favorable conditions for probiotic intake. As previously noted, LGG, a widely researched probiotic, has been shown to confer various health benefits, including enhancement of intestinal barrier function and modulation of the immune system [21]. In this study, two representative food matrices were selected: durum wheat pasta (a solid food) and soy milk (a liquid plant-based food). Pasta was chosen as a model solid matrix due to its popularity, ease of preparation, and good nutritional value including high carbohydrate and resistant starch content [22,23]. Soy milk, on the other hand, represents a widely consumed plant-based liquid alternative to dairy, with distinct protein and fat profiles [24]. This selection reflects globally prevalent dietary patterns and offers a contrast between complex carbohydrates and plant-based proteins, differing from prior studies that primarily focused on dairy or simple sugar-based matrices. Using a standardized static in vitro digestion model, we simulated the oral, gastric, and intestinal phases of digestion to assess LGG survival under three realistic consumption scenarios, i.e., pre-meal, with meal, or post-meal.

## 2. Materials and Methods

### 2.1. Probiotic Culture, Pasta, and Soy Milk

The LGG product was obtained from Culturelle^®^ (Amerifit, Inc., Cromwell, CT, USA) and had a declared concentration of 5.0 × 10^9^ colony-forming units (CFU)/g. Commercial spaghetti pasta (72 g total carbohydrates per 100 g) and soy milk (4.7 g protein per 250 mL) were purchased from a local market under the brands San Remo (Adelaide, Australia) and Australia’s Own (Noumi Pty Ltd., Ingleburn, NSW, Australia), respectively. The total starch content of the pasta was determined using a total starch assay kit (Neogen^®^, Megazyme Ltd., Lansing, MI, USA) and was 73.96 ± 2.91%. The detailed nutritional compositions of the durum wheat semolina pasta and soy milk, based on manufacturer information, are provided in Appendix A (Table A1 and Table A2).

### 2.2. Chemicals for Simulated Static In Vitro Digestion

Simulated digestion fluids, including simulated salivary fluid (SSF), simulated gastric fluid (SGF), and simulated intestinal fluid (SIF), were prepared according to a standardized method reported by Minekus, Alminger M Fau-Alvito [25]. An electrolyte stock solution was prepared using the following salts: KCl (0.5 mol/L), KH_2_PO_4_ (0.5 mol/L), NaHCO_3_ (1 mol/L), NaCl (2 mol/L), MgCl_2_(H_2_O)_6_ (0.15 mol/L), and (NH_4_)_2_CO_3_ (0.5 mol/L). The electrolyte stock solution and CaCl_2_(H_2_O)_2_ were sterilized by autoclaving for 15 min at 121 °C and pre-heated to 37 °C before use. The SSF was prepared by dissolving porcine pancreatic α-amylase (150 U/mL) and CaCl_2_(H_2_O)_2_ (0.15 mM) in the electrolyte stock solution at pH 7.0. The SGF was prepared by dissolving porcine pepsin (4000 U/mL) and CaCl_2_(H_2_O)_2_ (0.15 mM) in the electrolyte stock solution at pH 3.0. The SIF was prepared by dissolving porcine pancreatin (8000 USP U/mL, based on tryptic activity as specified by the supplier), CaCl_2_(H_2_O)_2_ (0.6 mM), and bile salt (20 mM) in the electrolyte stock solution at pH 7.0. The enzymes used were α-amylase from porcine pancreas (Neogen^®^, Megazyme Ltd., Lansing, MI, USA, E-PANAA), pepsin from porcine stomach (Chem-Supply Pty Ltd., Gillman, SA, Australia, PL082), pancreatin from porcine pancreas (Sigma-Aldrich, St. Luis, MO, USA, P7545, 8×USP), and porcine bile (Sigma-Aldrich, St. Louis, MO, USA, B8631). All the chemicals used were of analytical grade. The digestive enzymes were added to the electrolyte stock solution immediately prior to use in the digestion experiments.

### 2.3. In Vitro Static Digestion

The static in vitro simulated digestion experiment was conducted according to a standardized method with minor modifications [25]. A probiotic drink was prepared by dissolving 0.5 g of LGG probiotic powder in 10 mL of distilled water for each digestion experiment. This LGG-only solution was first subjected to simulated digestion as a control to preliminarily evaluate LGG survival. The static in vitro simulated digestion of cooked pasta consisted of oral, gastric, and intestinal phases, while soy milk and the probiotic drink underwent only the gastric and intestinal phase. Briefly, raw pasta samples were cooked in distilled water for 7 min and then cooled to approximately 40 °C prior to in vitro digestion. Then, 10 g of cooked pasta samples was weighed and broken down into smaller particles (approximately 2–5 mm) using a food processer (Magimix Cuisine 4200XL, Kitchen Warehouse, Victoria, Australia) to mimic oral mastication. The artificially masticated pasta was mixed with the SSF (37 °C) at a constant stirring speed of 100 rpm for 2 min to mimic oral digestion. The resulting mixture was mixed with the SGF at a volume ratio of 1:1 (*v*/*v*). For the soy milk, 10 mL of the sample was directly mixed with the SGF to achieve a sample-to-SGF ratio of 1:1 (*v*/*v*). The pH of both mixtures (pasta–SGF and soy milk–SGF) was adjusted to 3.0 using HCl, followed by incubation at 37 °C with shaking at 150 rpm for 2 h to simulate gastric digestion. Subsequently, 10 mL of the SIF was added to each sample to achieve a chyme: SIF ratio of 1:1 (*v*/*v*), and the pH was adjusted to 7.0 using NaOH. The intestinal digestion phase was carried out for 3 h under the same incubation conditions.

Probiotic administration was conducted by introducing the LGG probiotic drink either before, with, or after ingestion of the cooked pasta or soy milk, corresponding to three realistic mealtime scenarios. In the “pre-meal” group, the LGG drink was first mixed with the SGF and incubated at 37 °C for 30 min before the addition of the orally digested pasta or gastric-digested soy milk. In the “with meal” group, the LGG drink was added simultaneously with the cooked pasta at the oral phase or with the soy milk at the start of the gastric phase. Lastly, in the “post-meal” group, the LGG drink was added once the gastric digestion of cooked pasta or soy milk had processed for 30 min.

### 2.4. Microbiological Analysis

Samples for measuring the survival of the probiotics were taken at the following times: at the start (0 min) and end (2 min) of the oral phase; at the start of the gastric phase (0 min), mid-way through the gastric phase (60 min), and at the end of the gastric phase (120 min); at the start of the intestinal phase (0 min), at an earlier stage during the intestinal phase (30 min), at a later stage during the intestinal phase (60 min), and at the end of the intestinal phase (180 min). For the soy milk digestion, no samples were taken from the oral phase since it was excluded. This is consistent with the INFOGEST protocol’s recommendation to omit the oral phase for liquid foods that do not require mastication [25,26]. At each time point, 1 mL of sample was collected and centrifuged at 5000× *g* at 20 °C for 10 min, after which the resultant pellet was resuspended in 10 mL of 0.5% (*w*/*v*) peptone solution.

The enumeration of the LGG was carried out by the standard plate count method for microbiological analysis using a 0.5% (*w*/*v*) peptone solution as the diluent [27]. Briefly, 1 mL of the resuspended sample was added to 9 mL of a sterile peptone solution. The suspensions were then serially diluted 10-fold, followed by spreading 0.1 mL of the appropriate dilution on a de Man Rogosa Sharpe (MRS) agar plate for a 48 h stationary incubation at 37 °C. Plates containing 25–250 colonies were counted to calculate the number of viable cells in each sample. The moisture content of the samples was determined by drying in an oven at 80 °C for 48 h. Viability tests were carried out in duplicate, with each measurement performed in triplicate to obtain an average value.

### 2.5. pH Measurement

The pH of the chyme during the simulated digestion was measure and recorded at the start and end of the oral phase, and every 15 min from the start to the end of the gastric and intestinal phases. pH adjustments were made as necessary using HCl or NaOH based on these measurements to maintain physiological conditions.

### 2.6. Starch Hydrolysis of Pasta

The chyme samples collected from the pasta digestion were centrifuged at 5000 rpm for 10 min. A 100 μL aliquot of supernatant was taken and mixed with 1 mL of 0.5% (*w*/*v*) *p*-hydroxybenzoic acid hydrazide (PABAH) solution containing 0.5 M HCl and 0.5 M NaOH [28,29]. The mixture was incubated in boiling water for 5 min. After cooling, absorbance was measured at 410 nm using a spectrophotometer (Spectra Max M2, Molecular Devices, San Jose, CA, USA) to assess reducing sugar concentration as an indicator of starch hydrolysis.

### 2.7. Protein Digestibility of Soy Milk

The total protein concentration of the soy milk was determined using a bicinchoninic acid (BCA) protein assay kit (Thermo Fisher Scientific, Waltham, MA, USA). The chyme collected from the soy milk digestion was centrifuged at 12,000 rpm for 10 min to separate undigested protein (precipitate) from soluble protein (supernatant). The precipitate was then washed with distilled water and centrifuged again to remove residual soluble proteins. A 100 μL aliquot of the final supernatant was mixed with 1 mL of Tris HCl buffer. The working reagent was prepared by mixing reagent A and reagent B from the BCA kit at a ratio of 1:50. A standard curve was established using bovine serum albumin (BSA, Thermo scientific, Waltham, MA, USA). For quantification, 25 μL of either the BSA standards or the samples was pipetted into a microplate, followed by 200 μL of working reagent. The plate was shaken for 30 s and then incubated at 37 °C for 30 min, cooled to room temperature. The absorbance was measured at 562 nm using a Tecan Plate Reader (Infinite M200 Pro, Männedorf, Switzerland), and total protein content was quantified by comparing with the BSA standard. Protein digestibility was calculated as follows:(1)$$Protein\;digestibility\left(\%\right)=\left(\frac{Initial\;protein\;content-Undigested\;protein\;content}{Initial\;proteincontent}\right)\times 100\%$$

### 2.8. Statistical Analysis

All the results were expressed as mean ± standard deviation (SD). The experiments were performed in duplicate under each condition, and the control experiments without the LGG were conducted in parallel. Statistical analysis was performed using one-way analysis of variance (ANOVA) and Tukey’s test using SPSS version 3.0 (SPSS Inc., Chicago, IL, USA). Differences were considered significant at *p* < 0.05.

## 3. Results

### 3.1. Viability of LGG During Simulated In Vitro Static Digestion

The preliminary investigations into the gastrointestinal tolerance of LGG revealed distinct survival patterns across the different phases of static in vitro digestion (Figure 1). During the gastric phase, the LGG exhibited a moderate reduction in viability, with a loss of only 0.82 log CFU/g (a survival rate of 43.41%), indicating relatively good acid tolerance. However, the largest reduction in the viable cell concentration emerged when the LGG encountered the SIF containing pancreatin and bile salts, dropping from 9.06 to 6.53 log CFU/g (a survival rate of 0.0010%) (*p* < 0.05). This downward trend continued, reaching a final concentration of 4.73 log CFU/g by the end of the simulated digestion (a survival rate of 0.000016%).

### 3.2. Viability of LGG Co-Ingested with Food During Simulated Digestion

The protective effect of two different types of food matrices on the survival of LGG during simulated gastrointestinal digestion was investigated under three realistic consumption scenarios: (i) pre-meal (probiotic drink taken 30 min prior to food intake), (ii) with meal (probiotic drink co-consumed with food), and (iii) post-meal (probiotic drink administered 30 min after food consumption). The underlying hypothesis was that co-ingestion with food may buffer gastrointestinal stress and enhance probiotic survival.

Significant interactions were observed between the food matrix and LGG viability (Figure 2). As expected, co-ingestion with food improved the survival of the LGG compared to administration on an empty stomach. The durum wheat pasta provided the most robust protective effect, with final LGG counts reaching 6.04 ± 0.04 log CFU/g (a survival rate of 0.00033% in the pre-meal group), 5.92 ± 0.02 log CFU/g (a survival rate of 0.00025% in the with-meal group), and 6.38 ± 0.04 log CFU/g (a survival rate of 0.00072% in the post-meal group). These values represent a marked enhancement over the no-food control group. The soy milk also conferred a protective effect, albeit to a lesser extent. Final viable counts of LGG were 4.93 ± 0.00 (a survival rate of 0.000026%), 5.39 ± 0.01 (a survival rate of 0.000074%), and 5.19 ± 0.03 log (a survival rate of 0.000074%) CFU/g for the pre-meal, with-meal, and post-meal groups, respectively. While all the values were significantly higher than the probiotic-alone condition, the magnitude of protection was lower than that offered by the pasta. Temporal differences in probiotic administration (pre-, with, or post-meal) within each food matrix group did not reach statistical significance (*p* > 0.05). However, subtle trends were observed: the post-meal pasta and with-meal soy milk conditions yielded the highest LGG viability within their respective groups. Conversely, both matrices showed the lowest protective effects in the pre-meal group, suggesting that food presence during gastric passage is a key determinant in maintaining probiotic viability.

### 3.3. pH, Starch Hydrolysis, and Protein Digestibility

Figure 3 illustrates the pH profiles of the pasta and soy milk digestion. Both matrices exhibited similar pH patterns across the three digestive phases. The oral phase maintained a near-neutral environment (pH 7.01–7.03, observed only in the pasta “with meal” group), followed by acidification during the gastric phase (pH 2.78–3.00) and re-neutralization in the intestinal phase (pH 6.84–6.94).

Notably, the co-ingestion of the LGG probiotics modulated carbohydrate metabolism during digestion. Although probiotic administration appeared to delay the onset of starch hydrolysis, it ultimately enhanced the total starch digestibility in the pasta from 84.80 ± 0.47% (control) to 89.00 ± 1.00% with LGG supplementation. As depicted in Figure 4a, the most pronounced improvement was observed in the post-meal group, where the LGG was introduced 30 min after pasta ingestion.

In addition, protein digestibility in the soy milk also demonstrated a modest yet consistent enhancement in response to LGG supplementation. The final protein digestibility increased from 78.00 ± 1.00% to 80.00 ± 1.00% (Figure 4b). However, unlike the results for starch digestion, the timing of probiotic administration had a less discernible effect on protein digestion.

### 3.4. Impact of Different Food Matrices

The protective capacity of different food matrices on LGG viability during simulated digestion was further evaluated by comparing co-ingestion with pasta and soy milk. As illustrated in Figure 5, the pasta consistently provided a significantly stronger protective effect across all administration regimens (pre-meal, with meal, and post-meal), maintaining a notably higher viable cell concentration of LGG throughout the 3 h digestion period (*p* < 0.05). At the end of the simulated digestion, the LGG co-ingested with pasta retained 1.19–1.65 log CFU/g more viable cells compared to the baseline viability observed under fasting conditions (4.73 log CFU/g). In contrast, the soy milk resulted in a comparatively smaller viability enhancement, ranging from 0.20 to 0.66 log CFU/g, depending on administration timing. These findings clearly demonstrate the superior protective role of the pasta in preserving probiotic viability under gastrointestinal stress.

## 4. Discussion

One of the fundamental requirements for probiotic administration is ensuring that the probiotics survive transit through the GI tract and reach the colon in a viable state [7]. In this study, the substantial decline in LGG viability during the simulated digestion was consistent with previous reports on commercial probiotic products. For instance, da Silva, Tagliapietra [30] showed that all the samples tested experienced a reduction in viability ranging from 1 to 4 log CFU/g. Similarly, Treven, Paveljšek [18] found that co-digestion of probiotic products with water resulted in an average reduction of 1.6 log CFU/g in viability, with half of the tested samples showing a reduction between 1 and 4 log CFU/g. Aziz, Zaidi [31] observed that only approximately 22% of the tested products demonstrated post-gastrointestinal viability, while Millette, Nguyen [32] reported that 45% of the examined samples experienced a 1–5 log CFU/g reduction in viability after simulated digestion, with 31% showing complete loss of viability. In the current study, the lowest LGG survival was associated with exposure to pancreatin and bile salts, rather than gastric acidity or proteolytic enzymes. Bile salts, as anionic surfactants, disrupt bacterial membranes by solubilizing lipids and membrane proteins, posing a critical challenge to probiotic survival [33]. This observation supports previous conclusions that bile tolerance is a more decisive factor than acid resistance for probiotic functionality, especially since acid stress can often be mitigated through encapsulation or food co-ingestion [31,34].

Although current clinical studies have not established a definitive dose-response relationship for probiotic efficacy [35], there is a consensus that a minimum of 10^6^–10^7^ CFU/g of live probiotic cells must arrive at the target site of the GI tract to benefit the host [32]. Under fasting conditions, the LGG survival of 4.73 log CFU/g observed in this study fell below this threshold, highlighting the suboptimal viability of probiotics when consumed on an empty stomach. In contrast, co-ingestion with food significantly improved LGG survival, emphasizing the critical role of dietary context in enhancing probiotic resilience during digestion. Notably, pre-meal administration consistently resulted in the lowest survival rates for both food matrices, corroborating prior findings that discourage probiotic intake on an empty stomach. Similarly, Treven, Paveljšek [18] reported improved survival rates when products were co-digested with food. These differences in the viability coincide with prior findings, confirming that the timing of probiotic supplement intake relative to food exerts an impact on their survival [3,18].

One explanation for these results might be the interactions between probiotic cells and the surrounding food matrix. In this study, co-ingestion with LGG delayed starch hydrolysis but enhanced both starch and protein digestibility. Normally, the highest rate of starch hydrolysis occurred during the initial 30 min phase of digestion [28]. However, starch digestibility was only about 20% during the first hour when co-ingested with the LGG. Moreover, both starch and protein reached a final digestibility of at least 80% at the end of the simulated digestion. These values are considerably higher than the approximate 50% digestibility reported for pasta and soy milk alone [28,36]. Our results demonstrate that LGG co-ingestion enhanced starch digestibility in pasta and protein digestibility in soy milk. This synergistic effect likely occurs through two primary mechanisms. Firstly, probiotic microorganisms, such as LGG, can enhance enzymatic hydrolysis that contributes to the breakdown of complex macronutrients. For instance, the elimination of α-amylase inhibitors and the reduction in their activity by probiotic fermentation may be responsible for increased starch digestibility, while enhanced proteinase activity in fermented foods facilitates the hydrolysis of proteins intro smaller peptides and amino acids [37,38]. This enzymatic action directly increases the digestibility of co-ingested carbohydrates and proteins, as observed in our study with significant improvements in the starch and protein digestibility. Secondly, fermentation or the activity of probiotics during digestion can reduce levels of compounds that inhibit nutrient absorption such as phytic acid, which is known to inhibit proteolytic activity [38]. The reduction of anti-nutritional factors during the digestive process facilitated by probiotics further improves the accessibility and utilization of nutrients. Therefore, co-ingestion of probiotic supplements with food may not only enhance probiotic viability and effectiveness but also improve the bioavailability and nutritional quality of co-ingested foods.

Additionally, this study compared the buffering capacity of pasta and soy milk and evaluated their effects on the viability of LGG during simulated digestion. The obtained results suggest that the tolerance of the LGG to GI tract stresses, particularly the intestine environment, were strongly dependent on the chemical composition and physical state of the ingested food. We propose that the gelatinized starch–protein network formed during the cooking of pasta acts as a physical barrier, entrapping probiotic cells and reducing their direct exposure to bile salts and pancreatin in the intestine, thereby enhancing survival. The combined structure of gelatinized starch and protein forms a denser and more ordered physical barrier that surrounds starch granules or other components such as probiotic cells, making them less accessible to digestive enzymes [39,40]. These results are consistent with previous studies showing that specific food matrices can confer enhanced protection to probiotics during both storage and simulated GI conditions [11,13]. For instance, Matouskova, Hoova [11] found that combining probiotic strains with protein- and sugar-rich foods was as an effective strategy for improving probiotic survival during GI digestion. Similarly, Treven, Paveljšek [18] observed a more significant reduction in viable probiotic counts when co-digested with juice (1.1 to 5.3 log CFU/g) compared to porridge (0.2 to 3.7 log CFU/g). Tompkins, Mainville [3] further demonstrated the importance of the meal type, ranking the protective effect of different matrices in descending order as follows: 1% fat milk > oatmeal–milk gruel > apple juice > spring water.

Buffering capacity, defined as the ability of a food product to resist pH changes, is a key factor in improving probiotic survival during GI passage [22,41]. This capacity notably affects gastric digestion by modulating the physicochemical breakdown of food [42]. Generally, gastric pH stabilizes around 2.5–3.5 under fasting conditions, potentially increasing to 4–6 due to the buffering capacity introduced by food intake [43]. The buffering capacity of a food matrix is dependent on its intrinsic characteristics, including nutrient composition (fat, protein, and sugar content), particle size, and total solids [41,42]. All of these can impact probiotic survival and functionality.

Dairy products, particularly milk, are widely considered favorable carriers for probiotics due to their high pH, buffering capacity, and fat content, which help mitigate the harsh gastric and duodenal environments [44]. However, such protective effects were not observed for soy milk in this study. Soy milk is plant-based and lactose-free. Lactose is a key fermentable carbohydrate source known to serve as a preferred energy source for *lactobacilli* including LGG, supporting their growth [45]. Moreover, by fermenting lactose into lactic acid and other metabolic byproducts [46], the bacteria create an acidic environment that they are well-suited to tolerate, enhancing their acid and bile tolerance during GI transit [11]. Therefore, the absence of lactose likely reduced the soy milk’s protective effect during digestion. Furthermore, the presence of inherent antinutritional factors in soybeans, such as trypsin inhibitors and phytates, may indirectly alter the digestive environment via the inhibition of proteolytic enzymes [37,38]. It is important to note that while probiotic fermentation can reduce the level of antinutritional factors [47], their presence in soy milk underscores a fundamental compositional difference from dairy matrices, which are naturally free of such compounds and provide a more tailored nutrient profile for probiotics.

As per the Nutrition Facts labels on the products (see Appendix A, Table A1 and Table A2), the pasta contained 72 g carbohydrates, 12.5 g protein, and 2 g fat per 100 g serving, while the soy milk provided 1 g carbohydrates, 4.7 g protein, and 5.8 g fat per 100 g serving. Although fat and proteins have been proposed to improve the viability of probiotic bacteria during digestion [3,16,17,41,42], our results showed that co-digestion with pasta provided better protection for probiotics against digestive enzymes and bile salts than co-digestion with soy milk. The superior performance of the pasta can be attributed to its solid nature, which may have offered extended protective capacity compared to the liquid form of the soy milk, allowing a greater proportion of LGG cells to reach the duodenum before encountering detrimental conditions [3]. Having less moisture and higher solid content, pasta likely offers stronger buffering capacity than soy milk. This is in agreement with previous findings proving that total solid content correlates positively with buffering capacity and probiotic protection against an unfavorable environment [16,42]. Therefore, the differences in food composition may explain the distinct buffering capacities observed and their resulting levels of bacterial protection. These findings suggest that probiotic microorganisms exhibit better survival when consumed with solid, carbohydrate-rich foods such as pasta, rather than solely with beverages like soy milk. However, it is important to note that the present study did not directly measure the buffering capacity of pasta or soy milk. While our interpretation is supported by survival outcomes and the existing literature, future work should incorporate quantitative titration assays to determine the precise buffering strength of different matrices, thereby providing mechanistic validation of the protective effects observed.

In addition, the static in vitro digestion model used in this study has limitations including lack of the ability to simulate dynamic physiological processes such as peristalsis, gastric emptying, and sieving for solid foods during digestion [48,49]. Future work should employ dynamic digestion models (e.g., TIM-1, SHIME, or DHSI-IV) that better simulate human GI conditions [50]. Although dairy matrices such as milk and yogurt are among the most common carriers for probiotics in commercial formulations, they were not included in this study. Our focus on pasta and soy milk was intended to highlight solid, carbohydrate-rich and liquid, plant-based protein-rich matrices as dairy-free alternatives, which is particularly relevant for consumers avoiding dairy. Future studies should therefore extend this approach to dairy products to broaden the applicability of the findings across a wider spectrum of probiotic delivery formats. Nevertheless, our findings therefore establish critical relationships between food matrix characteristics, administration timing, and probiotic viability. Since the existing literature concerning the optimal time for the consumption of probiotic supplements is inconclusive, and few clinical trials describe the specific administration approach for probiotics, this study significantly advances current understanding of dietary strategies for probiotic optimization, providing a foundation for future in vitro and ultimately clinical studies.

## 5. Conclusions

This study investigated how the administration timing of LGG relative to food intake, and the buffering capacity of different food matrices, influenced the survival of LGG during GI transit. The findings confirm that specific food matrices, particularly solid, carbohydrate-rich foods such as durum wheat pasta, can enhance the probiotic viability under the harsh conditions of the GI tract. In contrast, liquid matrices such as soy milk offered less protection due to their weaker buffering capacity and less favorable composition for probiotic survival. Furthermore, the importance of timing for probiotic administration was emphasized, demonstrating that consuming probiotic supplements with or after a meal led to substantially better survival outcomes compared to consumption on an empty stomach. These insights underscore the need for clear dietary guidelines regarding both the selection of food matrices and the timing of probiotic intake. Overall, this work contributes to a deeper understanding of how dietary context influences probiotic efficacy and offers practical recommendations to improve the success of probiotic supplementation in real-world settings.

## Figures and Tables

**Figure 1 foods-14-03076-f001:**
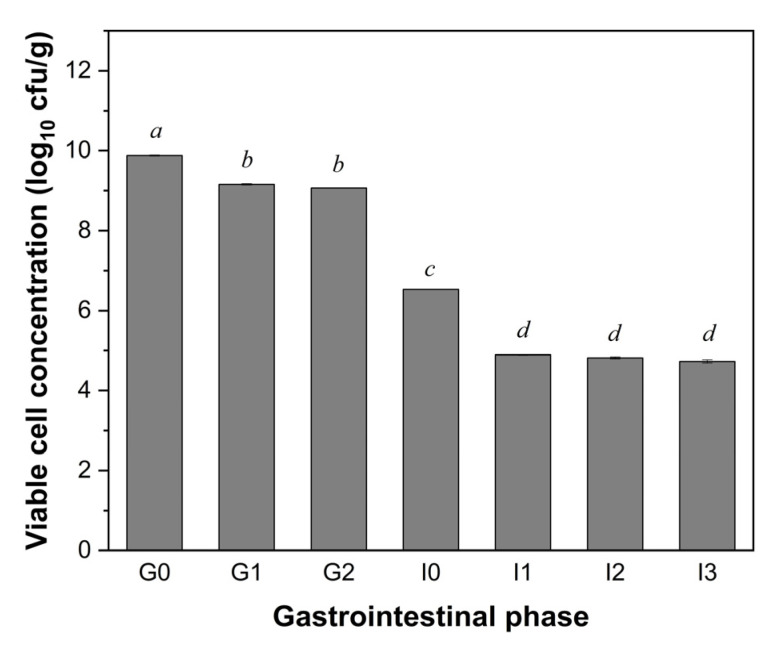
Changes in the viability (log CFU/g) of LGG at different stages during the simulated in vitro static digestion. G0: 0 min of the gastric phase; G1: 60 min of the gastric phase; G2: 120 min of the gastric phase; I0: 0 min of the intestinal phase; I1: 30 min of the intestinal phase; I2: 60 min of the intestinal phase; I3: 180 min of the intestinal phase. Values with different letters are significantly different (*p* < 0.05).

**Figure 2 foods-14-03076-f002:**
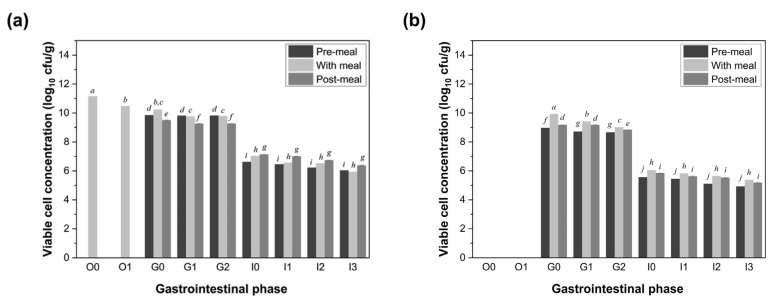
Changes in the viability (log CFU/g) of LGG at different stages during the simulated digestion of pasta (**a**) or soy milk (**b**) with the probiotic drink consumed under different scenarios. O0: 0 min of the oral phase; O1: 2 min of the oral phase; G0: 0 min of the gastric phase; G1: 60 min of the gastric phase; G2: 120 min of the gastric phase; I0: 0 min of the intestinal phase; I1: 30 min of the intestinal phase; I2: 60 min of the intestinal phase; I3: 180 min of the intestinal phase. Values with different letters (for pre-meal, with meal, or post-meal groups) are significantly different (*p* < 0.05).

**Figure 3 foods-14-03076-f003:**
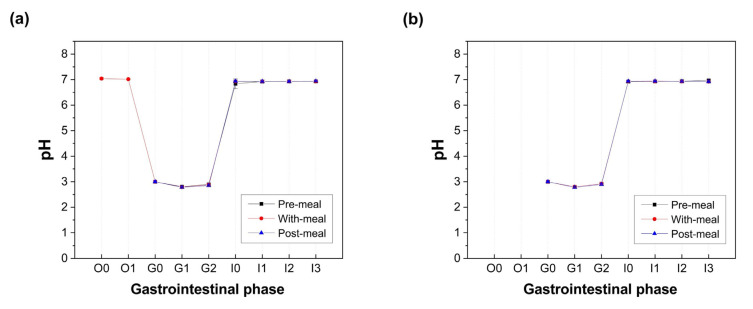
Changes in pH values of chyme at different stages during the simulated digestion of pasta (**a**) or soy milk (**b**) with the probiotic drink consumed under different scenarios. O0: 0 min of the oral phase; O1: 2 min of the oral phase; G0: 0 min of the gastric phase; G1: 60 min of the gastric phase; G2: 120 min of the gastric phase; I0: 0 min of the intestinal phase; I1: 30 min of the intestinal phase; I2: 60 min of the intestinal phase; I3: 180 min of the intestinal phase.

**Figure 4 foods-14-03076-f004:**
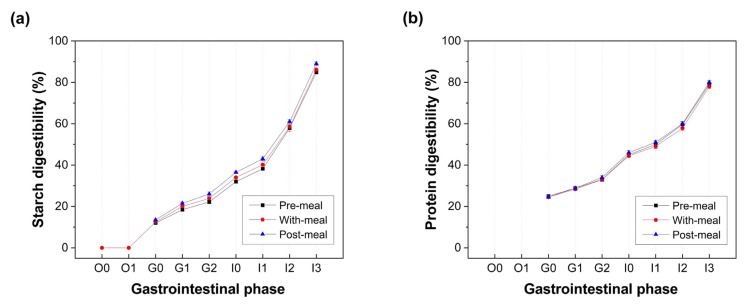
Starch digestibility of pasta (**a**) and protein digestibility of soy milk (**b**) at different stages during the simulated digestion of the probiotic drink consumed under different scenarios. O0: 0 min of the oral phase; O1: 2 min of the oral phase; G0: 0 min of the gastric phase; G1: 60 min of the gastric phase; G2: 120 min of the gastric phase; I0: 0 min of the intestinal phase; I1: 30 min of the intestinal phase; I2: 60 min of the intestinal phase; I3: 180 min of the intestinal phase.

**Figure 5 foods-14-03076-f005:**
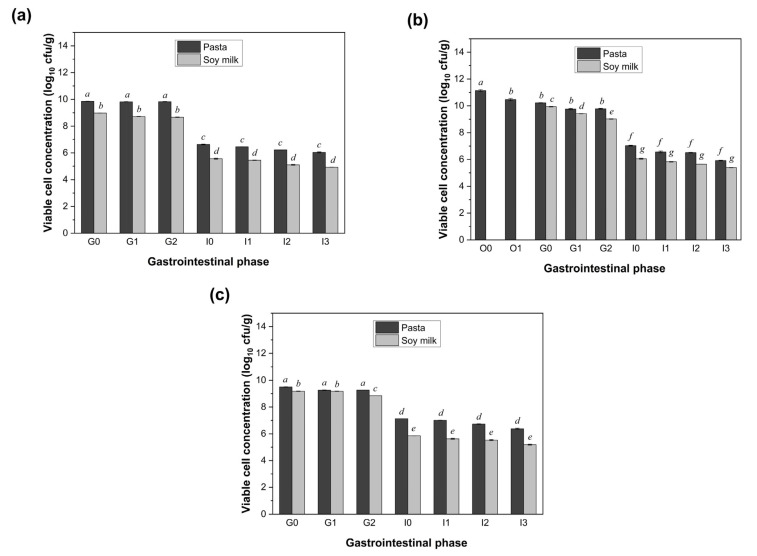
Viability of LGG at different stages during the simulated digestion of pasta and soy milk with the probiotic drink consumed before (**a**), during (**b**), or after (**c**) the meal. O0: 0 min of the oral phase; O1: 2 min of the oral phase; G0: 0 min of the gastric phase; G1: 60 min of the gastric phase; G2: 120 min of the gastric phase; I0: 0 min of the intestinal phase; I1: 30 min of the intestinal phase; I2: 60 min of the intestinal phase; I3: 180 min of the intestinal phase. Values with different letters (for either pasta or soy milk groups) are significantly different (*p* < 0.05).

## Data Availability

The original contributions presented in the study are included in the article. Further inquiries can be directed to the corresponding authors.

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
