# Peer review of "Effect of Food Matrix and Administration Timing on the Survival of Lactobacillus rhamnosus GG During In Vitro Gastrointestinal Digestion"

_foods, 2025, doi:10.3390/foods14173076_

Round 1
Reviewer 1 Report
Comments and Suggestions for Authors
-
Abstract: Add a quantifiable result to the abstract for immediate impact.
-
Introduction: The introduction can be improved by justifying the selection of durum wheat pasta and soy milk as food matrices. Any previous study can be cited as a review of the literature.
-
Materials and Methods: In Sections 2.2 (Chemicals for Simulated Static In Vitro Digestion) and 2.3 (In Vitro Static Digestion), specify the minor modifications made to the standard method.
- Formatting: The heading "2.5. pH Measurement" is repeated as "2.5. Starch Hydrolysis..." and "2.6. Protein Digestibility...". This is a minor formatting error.
- Line 110–111: Specify the concentration used: “Digestive enzymes were added to the electrolyte stock solution immediately prior to use in the digestion experiments.”
-
Line 146–147: “For soy milk digestion, no samples were taken from the oral phase since it was excluded.” The authors should briefly state why the oral phase was omitted for the liquid food, referring to the INFOGEST protocol.
-
Results Section: Calculate and report the percentage survival at each major phase (e.g., percentage survived gastric phase, percentage survived intestinal phase). This provides a more intuitive understanding of where the major damage occurs.
-
Discussion: The discussion can be improved by proposing a more detailed mechanistic model. It could focus on the dense, gelatinized starch network in pasta that acts as a physical barrier, entrapping probiotic cells and reducing their direct exposure to bile salts and pancreatin.
-
Discussion: Discuss why soy milk underperformed compared to dairy literature. Was it due to the lack of lactose or the presence of antinutritional factors?
-
Discussion: Beyond "buffering," elaborate on the physical mechanism. Propose that the dense, gelatinized starch network in pasta acts as a physical barrier, entrapping probiotic cells and reducing their direct exposure to bile salts and pancreatin. This could be a follow-up study using microscopy (e.g., SEM of digesta).
-
References: Update the references with recent ones: 2, 4, 6, 9, 10, 12, 25, 26, 27, 32, 33, 36, 37, 40, 42, 43, 45, 46.
-
References: References 3, 27, and 46 are incomplete. Format all references in a similar style.
Author Response
- Abstract:Add a quantifiable result to the abstract for immediate impact.
Response: We sincerely thank the reviewer for this valuable suggestion. As recommended, we have incorporated key quantifiable results into the abstract to provide immediate evidence of our findings. The updated abstract reads as follows:
Line 18-25: “…Pasta outperformed soy milk, yielding higher viable counts (5.92–6.38 vs. 4.93–5.39 log CFU/g) due to greater buffering capacity. Timing of administration also played a critical role: consuming probiotics with (5.39–5.92 log CFU/g) or after a meal (5.19–6.38 log CFU/g) enhanced viability compared to an empty-stomach scenario (4.93–6.04 log CFU/g). Additionally, LGG co-ingestion facilitated starch and protein digestion, increasing pasta starch digestibility from 84.80% to 89.00% and soy milk protein digestibility from 78.00% to 80.00%, suggesting synergistic bacteria-food interactions between the probiotic and food matrix.…”
- Introduction:The introduction can be improved by justifying the selection of durum wheat pasta and soy milk as food matrices. Any previous study can be cited as a review of the literature.
Response: We thank the reviewer for this insightful comment. Accordingly, we have revised the introduction to include references to previous studies and clearer reasoning for choosing these two specific matrices, emphasizing their global dietary relevance and compositional contrasts. The updated section in the introduction now reads as follows:
Line 80-83: “…Pasta was chosen as a model solid matrix due to its popularity, ease of preparation, and good nutritional value including high carbohydrate and resistant starch content [22, 23]. Soy milk, on the other hand, represents a widely consumed plant-based liquid alternative to dairy, with distinct protein and fat profiles [24]…”
- Materials and Methods:In Sections 2.2 (Chemicals for Simulated Static In Vitro Digestion) and 2.3 (In Vitro Static Digestion), specify the minor modifications made to the standard method.
Response: Thank you for raising this point. We aimed to provide a detailed and transparent description of the digestion protocol in our manuscript. Specifically, in Section 2.2, we explicitly listed all electrolytes, enzymes, and concentrations used in the simulated fluids, along with their sources and preparation methods. In Section 2.3, we described the digestion process step-by-step, including sample preparation, phase durations, pH adjustments, and timing of probiotic administration relative to meal phases. We believe these sections already contain a complete and reproducible account of our methodology.
- Formatting:The heading "2.5. pH Measurement" is repeated as "2.5. Starch Hydrolysis..." and "2.6. Protein Digestibility...". This is a minor formatting error.
Response: We sincerely thank the reviewer for careful reading and for identifying this numbering error in the Materials and Methods section. We have corrected the heading numbering in the revised manuscript, and the headings now correctly appear as: “… 2.5. pH Measurement … 2.6. Starch Hydrolysis of Pasta … 2.7. Protein Digestibility of Soy Milk … 2.8. Statistical Analysis…”
- Line 110–111:Specify the concentration used: “Digestive enzymes were added to the electrolyte stock solution immediately prior to use in the digestion experiments.”
Response: Thank you for this comment. We would like to clarify that the specific concentrations of the digestive enzymes used in each simulated fluid (SSF, SGF, and SIF) were already explicitly stated earlier in the manuscript in Section 2.2 (Lines 107, 108 and 110). Therefore, the sentence “Digestive enzymes were added to the electrolyte stock solution immediately prior to use in the digestion experiments,” serves as a general procedural statement that refers back to these previously specified concentrations. We believe that providing the concentrations again in this sentence would be redundant. We thank the reviewer for their meticulous reading and hope this clarification is satisfactory.
- Line 146–147:“For soy milk digestion, no samples were taken from the oral phase since it was excluded.” The authors should briefly state why the oral phase was omitted for the liquid food, referring to the INFOGEST protocol.
Response: Thank you for this comment. We have revised the sentence to include a brief justification for omitting the oral phase for liquid samples (soy milk), referencing the INFOGEST protocol's recommendations regarding the simulation of realistic consumption conditions for liquid foods. The updated text now reads as follows:
Line 156-158: “For soy milk digestion, no samples were taken from oral phase since it was excluded. This is consistent with the INFOGEST protocol's recommendation to omit the oral phase for liquid foods that do not require mastication [25, 26]…”
- Results Section:Calculate and report the percentage survival at each major phase (e.g., percentage survived gastric phase, percentage survived intestinal phase). This provides a more intuitive understanding of where the major damage occurs.
Response: We sincerely thank the reviewer for this insightful suggestion. We agree that reporting the percentage survival at each major digestive phase provides a more intuitive understanding of where the greatest viability losses occur. We have now calculated and incorporated these percentages into the Results (Section 3.1 and 3.2). The key revisions are as follows:
Line 208-214: “During the gastric phase, LGG exhibited a moderate reduction in viability, with a loss of only 0.82 log CFU/g (a survival rate of 43.41%), indicating relatively good acid tolerance. However, the largest reduction in the viable cell concentration emerged when LGG encountered SIF containing pancreatin and bile salts, dropping from 9.06 to 6.53 log CFU/g (a survival rate of 0.0010%) (p < 0.05). This downward trend continued, reaching a final concentration of 4.73 log CFU/g by the end of simulated digestion (a survival rate of 0.000016%).”
Line 230-238: “Durum wheat pasta provided the most robust protective effect, with final LGG counts reaching 6.04 ± 0.04 log CFU/g (a survival rate of 0.00033% in the pre-meal group), 5.92 ± 0.02 log CFU/g (a survival rate of 0.00025% in the with-meal group), and 6.38 ± 0.04 log CFU/g (a survival rate of 0.00072% in the post-meal group). These values represent a marked enhancement over the no-food control group. Soy milk also conferred a protective effect, albeit to a lesser extent. Final viable counts of LGG were 4.93 ± 0.00 (a survival rate of 0.000026%), 5.39 ± 0.01 (a survival rate of 0.000074%), and 5.19 ± 0.03 log (a survival rate of 0.000074%) CFU/g for the pre-meal, with-meal, and post-meal groups, respectively.”
- Discussion:The discussion can be improved by proposing a more detailed mechanistic model. It could focus on the dense, gelatinized starch network in pasta that acts as a physical barrier, entrapping probiotic cells and reducing their direct exposure to bile salts and pancreatin.
Response: We are very grateful to the reviewer for this insightful and valuable suggestion. We completely agree that proposing a more detailed mechanistic model would significantly strengthen the discussion of our results. We have substantially revised and expanded the Discussion (Section 4) to incorporate a focused explanation on the protective role of the gelatinized starch network in pasta, as proposed. The revised section in the Discussion now reads as follows:
Line 361-371: “The obtained results suggest that the tolerance of LGG to GI tract stresses, particularly the intestine environment, were strongly dependent on the chemical composition and physical state of the ingested food. We propose that the gelatinized starch-protein network formed during the cooking of pasta acts as a physical barrier, entrapping probiotic cells and reducing their direct exposure to bile salts and pancreatin in the intestine, thereby enhancing survival. The combined structure of gelatinized starch and protein forms a denser and more ordered physical barrier that surrounds starch granules or other components such as probiotic cells, making them less accessible to digestive enzymes [39, 40]. These results are consistent with previous studies showing that specific food matrices can confer enhanced protection to probiotics during both storage and simulated GI conditions [11, 13].”
- Discussion:Discuss why soy milk underperformed compared to dairy literature. Was it due to the lack of lactose or the presence of antinutritional factors?
Response: We sincerely thank the reviewer for raising this excellent point. Upon re-evaluating the literature, we agree that the role of antinutritional factors in soy milk regarding probiotic survival during digestion is complex and not directly documented. The primary reason for soy milk's underperformance is more robustly attributed to the lack of lactose, a crucial fermentable carbohydrate that supports probiotic growth and stress resistance. We have expanded the discussion on the mechanisms behind soy milk's underperformance to explicitly address the potential reasons. The revised section in the Discussion now reads as follows:
Line 387-401: “Dairy products, particularly milk, are widely considered favorable carriers for probiotics due to their high pH, buffering capacity, and fat content, which help mitigate the harsh gastric and duodenal environments [44]. However, such protective effects were not observed for soy milk in this study. Soy milk is plant-based and lactose-free. Lactose is a key fermentable carbohydrate source known to serve as a preferred energy source for lactobacilli including LGG, supporting their growth [45]. Moreover, by fermenting lactose into lactic acid and other metabolic byproducts [46], the bacteria create an acidic environment that they are well-suited to tolerate, enhancing their acid and bile tolerance during GI transit [11]. Therefore, the absence of lactose likely reduced soy milk’s protective effect during digestion. Furthermore, the presence of inherent antinutritional factors in soybeans, such as trypsin inhibitors and phytates, may indirectly alter the digestive environment by the inhibition of proteolytic enzymes [37, 38]. It is important to note that while probiotic fermentation can reduce the level of antinutritional factors [47], their presence in soy milk underscores a fundamental compositional difference from dairy matrices, which are naturally free of such compounds and provide a more tailored nutrient profile for probiotics.”
- Discussion:Beyond "buffering," elaborate on the physical mechanism. Propose that the dense, gelatinized starch network in pasta acts as a physical barrier, entrapping probiotic cells and reducing their direct exposure to bile salts and pancreatin. This could be a follow-up study using microscopy (e.g., SEM of digesta).
Response: We sincerely thank the reviewer for this insightful and constructive comment. We fully agree that going beyond the general concept of "buffering" and proposing a more detailed mechanistic explanation greatly enhances the scientific rigor and impact of our Discussion. In response, we have substantially revised Section 4 to explicitly describe the potential role of the gelatinized starch–protein network in pasta as a protective barrier. Specifically, we propose that this dense and ordered structure entraps probiotic cells, thereby limiting their direct exposure to bile salts and pancreatin in the intestinal phase and ultimately enhancing survival. This explanation integrates well with our findings and aligns with prior evidence that food matrices can protect probiotics by providing both chemical and structural barriers. The revised section of the Discussion now reads as follows:
Line 361-371: “The obtained results suggest that the tolerance of LGG to GI tract stresses, particularly the intestine environment, were strongly dependent on the chemical composition and physical state of the ingested food. We propose that the gelatinized starch-protein network formed during the cooking of pasta acts as a physical barrier, entrapping probiotic cells and reducing their direct exposure to bile salts and pancreatin in the intestine, thereby enhancing survival. The combined structure of gelatinized starch and protein forms a denser and more ordered physical barrier that surrounds starch granules or other components such as probiotic cells, making them less accessible to digestive enzymes [39, 40]. These results are consistent with previous studies showing that specific food matrices can confer enhanced protection to probiotics during both storage and simulated GI conditions [11, 13].”
- References:Update the references with recent ones: 2, 4, 6, 9, 10, 12, 25, 26, 27, 32, 33, 36, 37, 40, 42, 43, 45, 46.
Response: Thank you for this comment. We have carefully reviewed the entire reference list and have updated it by replacing several older references with more recent, relevant, and authoritative publications, except seminal works that remain highly relevant and authoritative in the field (e.g., the FAO/WHO guidelines on probiotics and the INFOGEST consensus method). The updates references are marked in red in the reference list.
- References:References 3, 27, and 46 are incomplete. Format all references in a similar style.
Response: Thank you for this comment. We sincerely apologize for these oversights. We have now carefully reviewed and corrected the entire reference list to ensure that all references are complete and formatted in a consistent style according to the journal's guidelines.

Reviewer 2 Report
Comments and Suggestions for Authors
This manuscript addresses a timely and practical question in probiotic research: how the choice of food carrier and the timing of administration affect the survival of Lactobacillus rhamnosus GG during gastrointestinal transit. By coupling a standardized static in vitro digestion model with simultaneous measurements of bacterial viability and macronutrient digestibility, the study provides new insights into the reciprocal interactions between probiotic cells and their dietary matrices. The clear demonstration that durum wheat pasta—especially when consumed with or shortly after probiotics—provides superior protection compared to soy milk highlights the importance of buffering capacity and food structure in probiotic delivery.
However, the central claim regarding the increased buffering capacity of pasta would benefit from quantitative validation. Simple acid-base titrations to determine the amount of standardized titrant required to shift each matrix by one pH unit would transform this claim into a rigorously supported conclusion. Such data could be presented in a concise table or accompanying figure, thereby reinforcing the mechanistic link between matrix chemistry and probiotic survival.
Although static in vitro systems are essential for initial screening, they cannot replicate dynamic processes such as peristalsis, gradual enzyme secretion, and fluctuating lumen pH. Acknowledging these limitations in the Discussion, along with a brief plan for dynamic simulation studies (e.g., TIM-1 or SHIME) or targeted in vivo validation, would clarify the scope and future directions of the study.
The manuscript’s focus on pasta and soy milk provides a compelling contrast between solid, carbohydrate-rich and liquid, protein-rich carriers. However, many commercial probiotic formulations rely on dairy matrices. A rationale for excluding dairy products—or an explicit proposal to investigate yogurt and milk in follow-up work—would extend the applicability of these findings to the diverse range of probiotic delivery formats on the market. Similarly, positioning the selected dose of 0.5 g LGG powder (≈2.5 × 10⁹ CFU) in the context of typical supplementation regimens (often ≥10⁹⁰ CFU) would help readers assess real-world relevance and potential dose-response effects.
The readability of the figures could be improved through larger fonts, different line styles or symbols for each administration scenario, and a simplified bar chart summarizing the final viability figures for immediate visual comparison.
The observation that co-ingestion of LGG improves the digestibility of starch and protein is intriguing and of broad nutritional interest. Basing this finding on established biochemical mechanisms—such as microbial amylase/protease activity or the degradation of antinutrients such as phytic acid—would add depth to the discussion and align the results with existing fermentation research.
Finally, minor methodological refinements (e.g., specifying the bile salt composition of intestinal fluid, correcting duplicate section numbering, and harmonizing CFU units between solid and liquid matrices), along with the addition of a concise nutritional composition table for the test foods and complete pH curves in the appendix would further enhance reproducibility.
Overall, this work makes a valuable contribution to evidence-based probiotic administration strategies. Only minor contextual clarifications are needed.
Author Response
This manuscript addresses a timely and practical question in probiotic research: how the choice of food carrier and the timing of administration affect the survival of Lactobacillus rhamnosus GG during gastrointestinal transit. By coupling a standardized static in vitro digestion model with simultaneous measurements of bacterial viability and macronutrient digestibility, the study provides new insights into the reciprocal interactions between probiotic cells and their dietary matrices. The clear demonstration that durum wheat pasta—especially when consumed with or shortly after probiotics—provides superior protection compared to soy milk highlights the importance of buffering capacity and food structure in probiotic delivery.
Response: We sincerely thank Reviewer #2 for thorough and constructive feedback, as well as for the overall positive assessment of our work. We agree with all the points raised and have revised the manuscript accordingly to address each suggestion. The changes are highlighted in the revised manuscript, and a point-by-point response is provided below.
However, the central claim regarding the increased buffering capacity of pasta would benefit from quantitative validation. Simple acid-base titrations to determine the amount of standardized titrant required to shift each matrix by one pH unit would transform this claim into a rigorously supported conclusion. Such data could be presented in a concise table or accompanying figure, thereby reinforcing the mechanistic link between matrix chemistry and probiotic survival.
Response: We sincerely thank the reviewer for this valuable suggestion. We agree that quantitative assessment of buffering capacity (e.g., by acid–base titration) would provide strong mechanistic support for our interpretation. Unfortunately, such measurements were not conducted in the present study, which is a limitation we now explicitly acknowledge in the revised manuscript. Instead, we based our interpretation on the observed survival differences and on previously reported buffering effects of cereal-based matrices. We have added a statement in the Discussion section highlighting the need for future studies to perform direct quantification of buffering capacity to further validate the protective role of pasta:
Line 419-423: “However, it is important to note that the present study did not directly measure the buffering capacity of pasta or soy milk. While our interpretation is supported by survival outcomes and existing literature, future work should incorporate quantitative titration assays to determine the precise buffering strength of different matrices, thereby providing mechanistic validation of the protective effects observed.”.
Although static in vitro systems are essential for initial screening, they cannot replicate dynamic processes such as peristalsis, gradual enzyme secretion, and fluctuating lumen pH. Acknowledging these limitations in the Discussion, along with a brief plan for dynamic simulation studies (e.g., TIM-1 or SHIME) or targeted in vivo validation, would clarify the scope and future directions of the study.
Response: We sincerely thank the reviewer for this critical and constructive suggestion. We fully agree that acknowledging the limitations of the static in vitro model is essential for defining future research directions. We have revised the Discussion section to explicitly address this point and to propose the use of advanced dynamic models as the logical next step. The revised section in the Discussion now reads as follows:
Line 424-440: “In addition, the static in vitro digestion model used in this study has limitations including lack of the ability to simulate dynamic physiological processes such as peristalsis, gastric emptying, and sieving for solid foods during digestion [48, 49]. Future work should employ dynamic digestion models (e.g., TIM-1, SHIME, or DHSI-Ⅳ) that better simulate human GI conditions [50]. Although dairy matrices such as milk and yogurt are among the most common carriers for probiotics in commercial formulations, they were not included in this study. Our focus on pasta and soy milk was intended to highlight solid, carbohydrate-rich and liquid, plant-based protein-rich matrices as dairy-free alternatives, which is particularly relevant for consumers avoiding dairy. Future studies should therefore extend this approach to dairy products to broaden the applicability of the findings across a wider spectrum of probiotic delivery formats. Nevertheless, our findings therefore establish critical relationships between food matrix characteristics, administration timing, and probiotic viability. Since existing literature concerning the optimal time for the consumption of probiotic supplements is inconclusive, and few clinical trials describe the specific administration approach for probiotics, this study significantly advances current understanding of dietary strategies for probiotic optimization, providing a foundation for future in vitro and ultimately clinical studies.”
The manuscript’s focus on pasta and soy milk provides a compelling contrast between solid, carbohydrate-rich and liquid, protein-rich carriers. However, many commercial probiotic formulations rely on dairy matrices. A rationale for excluding dairy products—or an explicit proposal to investigate yogurt and milk in follow-up work—would extend the applicability of these findings to the diverse range of probiotic delivery formats on the market. Similarly, positioning the selected dose of 0.5 g LGG powder (≈2.5 × 10⁹ CFU) in the context of typical supplementation regimens (often ≥10⁹⁰ CFU) would help readers assess real-world relevance and potential dose-response effects.
Response: We thank the reviewer for this insightful comment. While dairy matrices are indeed widely used in commercial probiotic products, our study intentionally focused on pasta and soy milk to investigate non-dairy carriers with contrasting nutritional and structural characteristics. Pasta, as a solid, carbohydrate-rich matrix, and soy milk, as a liquid, plant-based protein-rich matrix, provide valuable insights into alternative food vehicles beyond traditional dairy. This choice was also guided by the fact that the commercial probiotic formulation used in this study (Culturelle® LGG Kids powder) is dairy-free, making its combination with non-dairy matrices particularly relevant for consumers seeking dairy-free options. At the same time, we agree that extending this work to dairy products such as milk and yogurt would broaden the applicability of the findings to the dominant probiotic delivery formats in the market. Accordingly, we have revised the Introduction and Discussion to clarify our rationale for selecting pasta and soy milk, while also noting the importance of future work on dairy matrices:
Line 80-83: “…Pasta was chosen as a model solid matrix due to its popularity, ease of preparation, and good nutritional value including high carbohydrate and resistant starch content [22, 23]. Soy milk, on the other hand, represents a widely consumed plant-based liquid alternative to dairy, with distinct protein and fat profiles [24].…”
Line 429-435: “Although dairy matrices such as milk and yogurt are among the most common carriers for probiotics in commercial formulations, they were not included in this study. Our focus on pasta and soy milk was intended to highlight solid, carbohydrate-rich and liquid, plant-based protein-rich matrices as dairy-free alternatives, which is particularly relevant for consumers avoiding dairy. Future studies should therefore extend this approach to dairy products to broaden the applicability of the findings across a wider spectrum of probiotic delivery formats.”.
The readability of the figures could be improved through larger fonts, different line styles or symbols for each administration scenario, and a simplified bar chart summarizing the final viability figures for immediate visual comparison.
Response: We thank the reviewer for valuable feedback regarding the presentation of our figures. We agree that maximizing readability is important, and we have made every effort to optimize the clarity of the figures within the constraints of presenting complex, multi-timepoint data. In preparing the figures, we carefully balanced the need for clear presentation with the necessity of displaying all relevant data and trends for the different administration scenarios. For example, we have removed the data labels from the bar charts to reduce visual clutter and improve overall clarity; we have also ensured that all font sizes are maximized to the fullest extent possible while maintaining proper layout and proportion within each figure. We believe that the current presentation, supplemented with the detailed numerical data provided in the corresponding Results section, offers a clear representation of our findings. The figures were designed to balance comprehensive data presentation with visual accessibility, and we have made every effort to optimize both aspects.
The observation that co-ingestion of LGG improves the digestibility of starch and protein is intriguing and of broad nutritional interest. Basing this finding on established biochemical mechanisms—such as microbial amylase/protease activity or the degradation of antinutrients such as phytic acid—would add depth to the discussion and align the results with existing fermentation research.
Response: We appreciate the reviewer’s insightful comment. In our original Discussion section, we had already included a dedicated paragraph (Lines 342–356) that outlines the biochemical mechanisms underlying the observed effects. Specifically, we discussed (i) the role of probiotic microorganisms, such as LGG, in enhancing enzymatic hydrolysis of starch and protein through microbial amylase/protease activity and the reduction of α-amylase inhibitors, and (ii) the degradation of anti-nutritional factors such as phytic acid, which improves nutrient accessibility. This section highlights how these mechanisms collectively improve the digestibility of starch and protein, thereby supporting our finding that co-ingestion of LGG is beneficial. We believe this paragraph already incorporates the fundamental mechanisms suggested by the reviewer, and we have slightly refined the wording to make this mechanistic connection more explicit in the revised manuscript.
Line 342-359: “…This synergistic effect likely occurs through two primary mechanisms. Firstly, probiotic microorganisms, such as LGG, can enhance enzymatic hydrolysis that contribute to the breakdown of complex macronutrients. For instance, the elimination of α-amylase inhibitors and the reduction in their activity by probiotic fermentation may be responsible for increased starch digestibility, while enhanced proteinase activity in fermented foods facilitates the hydrolysis of proteins intro smaller peptides and amino acids [37, 38]. This enzymatic action directly increases the digestibility of co-ingested carbohydrates and proteins, as observed in our study with significant improvements in starch and protein digestibility. Secondly, fermentation or the activity of probiotics during digestion can reduce levels of compounds that inhibit nutrient absorption such as phytic acid, which is known to inhibit proteolytic activity [38]. The reduction of anti-nutritional factors during the digestive process facilitated by probiotics further improves the accessibility and utilization of nutrients. Therefore, co-ingestion of probiotic supplements with food may not only enhance probiotic viability and effectiveness, but also improve the bioavailability and nutritional quality of co-ingested foods….”.
Finally, minor methodological refinements (e.g., specifying the bile salt composition of intestinal fluid, correcting duplicate section numbering, and harmonizing CFU units between solid and liquid matrices), along with the addition of a concise nutritional composition table for the test foods and complete pH curves in the appendix would further enhance reproducibility.
Response: We sincerely thank the reviewer for the careful attention to methodological details, which has greatly improved the rigor and reproducibility of our manuscript. We have implemented all suggested refinements as outlined below:
- Bile Salt Composition: The precise type and concentration of bile salts have been added to Section 2.2. The revised text reads:
"... and bile salt (20 mM) in the electrolyte stock solution at pH 7.0. The enzymes used were α-amylase from porcine pancreatic (Neogen®, Megazyme Ltd, Lansing, MI, U.S.A., E-PANAA), pepsin from porcine stomach (Chem-Supply Pty Ltd, Gillman, SA, Australia, PL082), pancreatin from porcine pancreas (Sigma-Aldrich, St. Luis, MO, U.S.A., P7545, 8xUSP) and porcine bile (Sigma-Aldrich, St. Louis, MO, USA, B8631)." (Line 111-116)
- Duplicate Section Numbering: We have corrected the heading numbering in the revised manuscript, and the headings now correctly appear as:
“… 2.5. pH Measurement … 2.6. Starch Hydrolysis of Pasta … 2.7. Protein Digestibility of Soy Milk … 2.8. Statistical Analysis…”
- Harmonized CFU Units: All viability data for solid (pasta) and liquid (soy milk, probiotic drink) matrices have already been consistently reported as log CFU/g.
- Nutritional Composition Table: To improve transparency, we have added detailed nutritional composition tables for pasta and soy milk (based on manufacturer data) to the Appendix (Tables A1 and A2).
- pH Curves: Complete pH curves recorded during digestion have already been included in Figure 3 to further support reproducibility and allow readers to assess buffering behavior of each matrix.
Appendix Table A1. Nutritional composition of durum wheat semolina pasta (per 100 g, as provided by manufacturer).
|
Component |
Quantity per 100 g |
|
Energy |
|
|
‒ kJ |
1530 kJ |
|
‒ kcal |
366 kcal |
|
Protein |
12.5 g |
|
Fat, total |
2.0 g |
|
‒ Trans |
<1 g |
|
‒ Saturated |
<1 g |
|
‒ Polyunsaturated |
1.2 g |
|
‒ Monounsaturated |
<1 g |
|
Carbohydrates |
72.0 g |
|
‒ Sugars |
2.5 g |
|
Dietary Fibre |
3.0 g |
|
Sodium |
30 mg |
Appendix Table A2. Nutritional composition of soy milk (per 250 mL, as provided by manufacturer).
|
Component |
Quantity per 100 mL |
|
Energy |
|
|
‒ kJ |
134 kJ |
|
‒ kcal |
32 kcal |
|
Protein |
1.9 g |
|
Fat, total |
2.3 g |
|
‒ Trans |
<0.6 g |
|
‒ Saturated |
0.3 g |
|
‒ Polyunsaturated |
0.8 g |
|
‒ Monounsaturated |
1.2 g |
|
Carbohydrates |
0.4 g |
|
‒ Sugars |
0.4 g |
|
‒ Lactose |
0.0 g |
|
‒ Galactose |
0.0 g |
|
Dietary Fibre |
<0.5 g |
|
Sodium |
70 mg |
Overall, this work makes a valuable contribution to evidence-based probiotic administration strategies. Only minor contextual clarifications are needed.
Response: Thank you again for your thorough and constructive feedback.

Reviewer 3 Report
Comments and Suggestions for Authors
Comments to authors
Thank you for allowing me to participate in the review of the manuscript entitled “Effect of food matrix and administration timing on the survival of Lactobacillus rhamnosus GG during in vitro gastrointestinal digestion.” The manuscript addresses an important and timely question: how food matrices and administration timing influence the survival of L. rhamnosus GG (LGG) during simulated gastrointestinal (GI) digestion. The topic is highly relevant to both food science and nutrition, as probiotic efficacy depends strongly on survival through the GI tract. The study is generally well-designed, employs a standardized in vitro digestion model, and provides practical insights that could inform dietary guidelines for probiotic consumption. It is recommended to resolve the following comment:
Abstract: Clearly mention that this was an in vitro study to avoid misleading readers into assuming human or animal trials.
Introduction: Some redundancy (e.g., probiotics’ poor survival in the GI tract) could be condensed.
Line 106-109: The enzymes used…..bovine bile (Sigma-Aldrich, St. Louis, MO, USA). Please rewrite for clarity.
Line 115: Please correct “distil” water.
Line 205: “G0: 0 min of the oral phase”. Is this 0 min of the gastric phase? Please confirm and correct throughout the manuscript.
Discussion: Generally clear, but some sentences in the Discussion are long and repetitive. Concise writing would improve readability.
Minor typographical errors (e.g., “ProteinDigestibility” without spacing, and inconsistent in reference formatting).
Author Response
Thank you for allowing me to participate in the review of the manuscript entitled “Effect of food matrix and administration timing on the survival of Lactobacillus rhamnosus GG during in vitro gastrointestinal digestion.” The manuscript addresses an important and timely question: how food matrices and administration timing influence the survival of L. rhamnosus GG (LGG) during simulated gastrointestinal (GI) digestion. The topic is highly relevant to both food science and nutrition, as probiotic efficacy depends strongly on survival through the GI tract. The study is generally well-designed, employs a standardized in vitro digestion model, and provides practical insights that could inform dietary guidelines for probiotic consumption. Therefore, it is recommended for publication in foods after resolving the following comment:
Response: We sincerely thank you for thorough and constructive feedback, as well as for the overall positive assessment of our work. We agree with all the points raised and have revised the manuscript accordingly to address each suggestion. The changes are highlighted in the revised manuscript, and a point-by-point response is provided below.
Abstract: Clearly mention that this was an in vitro study to avoid misleading readers into assuming human or animal trials.
Response: We thank the reviewer for this important observation. We have revised the abstract to explicitly mention that this was an in vitro study. The revised text now reads as follows:
Line 14-17: “…Addressing the lack of dietary guidelines for optimal probiotic administration, this in vitro study examines how consuming Lactobacillus rhamnosus GG (LGG) with different foods at varying timings affects bacterial survival during simulated digestion.…”
Introduction: Some redundancy (e.g., probiotics’ poor survival in the GI tract) could be condensed.
Response: We thank the reviewer for this helpful suggestion. We have carefully reviewed the section and condensed the content to eliminate redundancy, while preserving the essential background information and logical flow of the argument.
Line 106-109: The enzymes used…..bovine bile (Sigma-Aldrich, St. Louis, MO, USA). Please rewrite for clarity.
Response: We have rewritten this sentence for improved clarity. The text now reads: "... and bile salt (20 mM) in the electrolyte stock solution at pH 7.0. The enzymes used were α-amylase from porcine pancreatic (Neogen®, Megazyme Ltd, Lansing, MI, U.S.A., E-PANAA), pepsin from porcine stomach (Chem-Supply Pty Ltd, Gillman, SA, Australia, PL082), pancreatin from porcine pancreas (Sigma-Aldrich, St. Luis, MO, U.S.A., P7545, 8xUSP) and porcine bile (Sigma-Aldrich, St. Louis, MO, USA, B8631)." (Line 111-116)
Line 115: Please correct “distil” water.
Response: We apologize for this typographical error. It has been corrected.
Line 205: “G0: 0 min of the oral phase”. Is this 0 min of the gastric phase? Please confirm and correct throughout the manuscript.
Response: We thank the reviewer for catching this significant labeling error. "G0" indeed refers to the start of the gastric phase, not the oral phase. This error has been corrected consistently throughout the entire manuscript.
Discussion: Generally clear, but some sentences in the Discussion are long and repetitive. Concise writing would improve readability.
Response: We thank the reviewer for this helpful suggestion. We have carefully reviewed the section and condensed the content to eliminate redundancy, while preserving the essential background information and logical flow of the argument.
Minor typographical errors (e.g., “ProteinDigestibility” without spacing, and inconsistent in reference formatting).
Response: We appreciate the reviewer's attention to detail. We have performed a complete check of the manuscript to correct these and other minor errors, and have now carefully reviewed and corrected the entire reference list to ensure that all references are complete and formatted in a consistent style according to the journal's guidelin
